# Learning Multiple Models via Regularized Weighting

**Daniel Vainsencher**
Department of Electrical Engineering
Technion, Haifa, Israel
danielv@tx.technion.ac.il

**Shie Mannor**
Department of Electrical Engineering
Technion, Haifa, Israel
shie@ee.technion.ac.il

**Huan Xu**
Mechanical Engineering Department
National University of Singapore, Singapore
mpexuh@nus.edu.sg

## Abstract

We consider the general problem of Multiple Model Learning (MML) from data, from the statistical and algorithmic perspectives; this problem includes clustering, multiple regression and subspace clustering as special cases. A common approach to solving new MML problems is to generalize Lloyd's algorithm for clustering (or Expectation-Maximization for soft clustering). However this approach is unfortunately sensitive to outliers and large noise: a single exceptional point may take over one of the models.

We propose a different general formulation that seeks for each model a distribution over data points; the weights are regularized to be sufficiently spread out. This enhances robustness by making assumptions on class balance. We further provide generalization bounds and explain how the new iterations may be computed efficiently. We demonstrate the robustness benefits of our approach with some experimental results and prove for the important case of clustering that our approach has a non-trivial breakdown point, i.e., is guaranteed to be robust to a fixed percentage of adversarial unbounded outliers.

## 1 Introduction

The standard approach to learning models from data assumes that the data were generated by a certain model, and the goal of learning is to recover this generative model. For example, in linear regression, an unknown linear functional, which we want to recover, is believed to have generated covariate-response pairs. Similarly, in principal component analysis, a random variable in some unknown low-dimensional subspace generated the observed data, and the goal is to recover this low-dimensional subspace. Yet, in practice, it is common to encounter data that were generated by a mixture of several models rather than a single one, and the goal is to learn a number of models such that any given data can be explained by at least one of the learned models. It is also common for the data to contain outliers: data-points that are not well explained by any of the models to be learned, possibly inserted by external processes.

We briefly explain our approach (presented in detail in the next section). At its center is the problem of assigning data points to models, with the main consideration that every model be consistent with many of the data points. Thus we seek for each model a distribution of weights over the data points, and encourage even weights by regularizing these distributions (hence our approach is called Regularized Weighting; abbreviated as RW). A data point that is inconsistent with all available models will receive lower weight and even sometimes be ignored. The value of ignoring difficult points is illustrated by contrast with the common approach, which we consider next.

The arguably most widely applied approach for multiple model learning is the minimum loss approach, also known as Lloyd's algorithm [1] in clustering, where the goal is to find a set of models, associate each data point to one model (in so called "soft" variations, one or more models), such that the sum of losses over data points is minimal. Notice that in this approach, every data point must be explained by some model. This leaves the minimum loss approach vulnerable to outliers and corruptions: If one data point goes to infinity, so must at least one model.

Our remedy to this is relaxing the requirement that each data point must be explained. Indeed, as we show later, the RW formulation is provably robust in the case of clustering, in the sense of having non-zero breakdown point [2]. Moreover, we also establish other desirable properties, both computational and statistical, of the proposed method. Our main contributions are:

1. A new formulation of the sub-task of associating data points to models as a convex optimization problem for setting weights. This problem favors broadly based models, and may ignore difficult data points entirely. We formalize such properties of optimal solutions through analysis of a strongly dual problem. The remaining results are characteristics of this approach.

2. Outlier robustness. We show that the breakdown point of the proposed method is bounded away from zero for the clustering case. The breakdown point is a concept from robust statistics: it is the fraction of adversarial outliers that an algorithm can sustain without having its output arbitrarily changed.

3. Robustness to fat tailed noise. We show, empirically on a synthetic and real world datasets, that our formulation is more resistant to fat tailed additive noise.

4. Generalization. Ignoring some of the data, in general, may lead to overfitting. We show that when the parameter $\alpha$ (defined in Section 2) is appropriately set, this essentially does not occur. We prove this through uniform convergence bounds resilient to the lack of efficient algorithms to find near-optimal solutions in multiple model learning.

5. Computational complexity. As almost every method to tackle the multiple model learning problem, we use alternating optimization of the models and the association (weights), i.e., we iteratively optimize one of them while fixing the other. Our formulation for optimizing the association requires solving a quadratic problem in $kn$ variables, where $k$ is the number of models and $n$ is the number of points. Compared to $O(kn)$ steps for some formulations, this seems expensive. We show how to take advantage of the special problem structure and repetition in the alternating optimization subproblems to reduce this cost.

## 1.1 Relation to previous work

Learning multiple models is by no means a new problem. Indeed, special examples of multi-model learning have been studied, including $k$-means clustering [3, 4, 5] (and many other variants thereof), Gaussian mixture models (and extensions) [6, 7] and subspace segmentation problem [8, 9, 10]; see Section 2 for details. Fewer studies attempt to cross problem type boundaries. A general treatment of the sample complexity of problems that can be interpreted as learning a code book (which encompasses some types of multiple model learning) is [11]. Slightly closer to our approach is [12], whose formulation generalizes a common approach to different model types and permits for problem specific regularization, giving both generalization results and algorithmic iteration complexity results. A probabilistic and generic algorithmic approach to learning multiple models is Expectation Maximization [13].

Algorithms for dealing with outliers and multiple models together have been proposed in the context of clustering [14]. Reference [15] provides an example of an algorithm for outlier resistance in learning a single subspace, and partly inspires the current work. In contrast, we abstract almost completely over the class of models, allowing both algorithms and analysis to be easily reused to address new classes.

## 2 Formulation

In this section we show how multi-model learning problems can be formed from simple estimation problem (where we seek to explain weighted data points by a single model), and imposing a par-

ticular *joint loss*. We contrast the joint loss proposed here to a common one through the weights assigned by each and their effects on robustness.

We refer throughout to $n$ data points from $\mathcal{X}$ by $(x_i)_{i=1}^n = X \in \mathcal{X}^n$, which we seek to explain by $k$ models from $\mathcal{M}$ denoted $(m_j)_{j=1}^k = M \in \mathcal{M}^k$. A data set may be weighted by a set of $k$ distributions $(\mathbf{w}_j)_{j=1}^k = W \in (\triangle^n)^k$ where $\triangle^n \subset \mathbb{R}^n$ is the simplex.

**Definition 1.** A *base weighted learning problem* is a tuple $(\mathcal{X}, \mathcal{M}, \ell, \mathcal{A})$, where $\ell : \mathcal{X} \times \mathcal{M} \to \mathbb{R}_+$ is a non-negative convex function, which we call a *base loss function* and $\mathcal{A} : \triangle^n \times \mathcal{X}^n \to \mathcal{M}$ defines an efficient algorithm for choosing a model. Given the weight $\mathbf{w}$ and data $X$, $\mathcal{A}$ obtains low weighted empirical loss $\sum_{i=1}^n \mathbf{w}_i \ell(x_i, m)$ (the weighted empirical loss need not be minimal, allowing for regularization which we do not discuss further).

We will often denote the losses of a model $m$ over $X$ as a vector $\mathbf{l} = (\ell(x_i, m))_{i=1}^n$. In the context of a set of models $M$, we similarly associate the loss vector $\mathbf{l}_j$ and the weight vector $\mathbf{w}_j$ with the model $m_j$; this allows us to use the terse notation $\mathbf{w}_j^\top \mathbf{l}_j$ for the weighted loss of model $j$.

Given a base weighted learning problem, one may pose a multi-model learning problem

**Example 1.** The multi-model learning problem covers many examples, here we list a few:

- In *k-means clustering*, the goal is to partition the training samples into $k$ subsets, where each subset of samples is "close" to their mean. In our terminology, a multi-model learning problem where the base learning problem is $\left(\mathbb{R}^d, \mathbb{R}^d, (x, m) \mapsto \|x - m\|_2^2, \mathcal{A}\right)$ where $\mathcal{A}$ finds the weighted mean of the data. The weights allow us to compute each cluster center according to the relevant subset of points.

- In *subspace clustering*, also known as subspace segmentation, the objective is to group the training samples into subsets, such that each subset can be well approximated by a low-dimensional affine subspace. This is a multi-model learning problem where the corresponding single-model learning problem is PCA.

- *Regression clustering* [16] extends the standard linear regression problem in that the training samples cannot be explained by one linear function. Instead, multiple linear function are sought, so that the training samples can be split into groups, and each group can be approximated by one linear function.

- *Gaussian Mixture Model* considers the case where data points are generated by a mixture of a finite number of Gaussian distributions, and seeks to estimate the mean and variance of each of these distribution, and simultaneously to group the data points according to the distribution that generates it. This is a multi-model learning problem where the respective single model learning problem is estimating the mean and variance of a distribution.

The most common way to tackle the multiple model learning problem is the minimum loss approach, i.e, to minimize the following joint loss

$$L(X, M) = \frac{1}{n} \sum_{x \in X} \min_{m \in M} \ell(x, m). \tag{2.1}$$

In terms of weighted base learning problems, each model gives equal weight to all points for which it is the best (lowest loss) model. For example, when $\mathcal{M} = \mathcal{X} = \mathbb{R}^n$ with $\ell(x, m) = \|x - m\|_2^2$ the squared Euclidean distance loss yields $k$ means clustering. In this context, alternating between choosing for each $x$ its loss minimizing model, and adjusting each model to minimized the squared Euclidean loss, yields Lloyd's algorithm (and its generalizations for other problems).

The minimum loss approach requires that *every* point is assigned to a model, this can potentially cause problems in the presence of outliers. For example, consider the clustering case where the data contain a *single* outlier point $x_i$. Let $x_i$ tend to infinity; there will always be some $m_j$ that is closest to $x_i$, and is therefore (at equilibrium) the average of $x_i$ and some other data points. Then $m_j$ will tend to infinity also. We call this phenomenon mode I of sensitivity to outliers; it is common also

to such simple estimators as the mean. Mode II of sensitivity is more particular: as $m_j$ follows $x_i$ to infinity, it stops being the closest to any other points, until the model is associated *only to the outlier* and thus matches it perfectly. Thus under Eq. (2.1) outliers tend to take over models. Mode II of sensitivity is not clustering specific, and Fig. 2.1 provides an example in multiple regression. Neither mode is avoided by spreading a point's weight among models as in mixture models [6].

To overcome both modes of sensitivity, we propose a different joint loss, in which the hard constraint is only that for *each model* we produce a distribution over *data points*. A penalty term discourages the concentration of a model on few points and thus mode II sensitivity. Deweighting difficult points helps mitigate mode I. For clustering this robustness is formalized in Theorem 2.

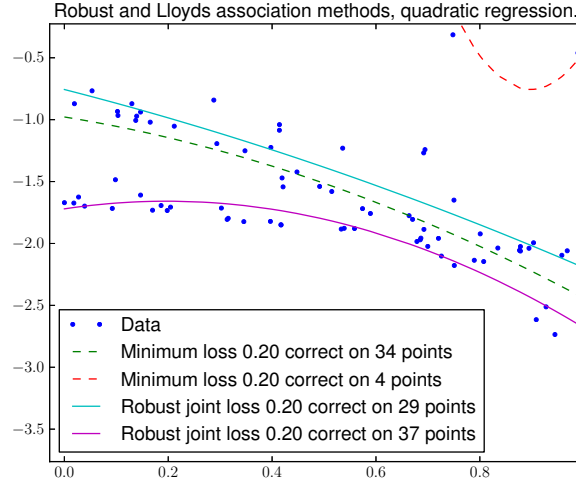

Figure 2.1: Data is a mixture of two quadratics, with positive fat tailed noise. Under a minimum loss approach an off-the-chart high-noise point suffices to prevent the top broken line from being close to many other data points. Our approach is free to better model the bulk of data. We used a robust (mean absolute deviation) criterion to choose among the results of multiple restarts for each model.

**Definition 2.** Let $\mathbf{u} \in \triangle^n$ be the uniform distribution. Given $k$ weight vectors, we denote their average $\mathbf{v}(W) = k^{-1} \sum_{j=1}^{k} \mathbf{w}_j$, and just $\mathbf{v}$ when $W$ is clear from context. The *Regularized Weighting* multiple model learning loss is a function $L_\alpha : \mathcal{X}^n \times \mathcal{M}^k \times (\triangle^n)^k \to \mathbb{R}$ defined as

$$L_\alpha(X, M, W) = \alpha \|\mathbf{u} - \mathbf{v}(W)\|_2^2 + k^{-1} \sum_{j=1}^{k} \mathbf{l}_j^\top \mathbf{w}_j \tag{2.2}$$

which in particular defines the *weight setting* subproblem:

$$L_\alpha(X, M) = \min_{W \in (\triangle^n)^k} L_\alpha(X, M, W). \tag{2.3}$$

As its name suggests, our formulation regularizes *distributions of weight over data points*; specifically, $\mathbf{w}_j$ are controlled by forcing their average $\mathbf{v}$ to be close to the uniform distribution $\mathbf{u}$. Our goal is for each model to represent many data points, so weights should not be concentrated. We avoid this by penalizing squared Euclidean distance from uniformity, which emphasizes points receiving weight much higher than the natural $n^{-1}$, and essentially ignores small variations around $n^{-1}$. The effect is later formalized in Lemma 1, but to illustrate we next calculate the penalties for two stylized cases. This will also produce the first of several hints about the appropriate range of values for $\alpha$.

In the following examples, we will consider a set of $\gamma n k^{-1}$ data points, recalling that $n k^{-1}$ is the natural number of points per model. To avoid letting a few high loss outliers skew our models (mode I of sensitivity), we prefer instead to give them zero weight. Take $\gamma \ll k/2$, then the cost of ignoring some $\gamma n k^{-1}$ points in all models is at most $\alpha n^{-1} \cdot 2\gamma k^{-1} \ll \alpha n^{-1}$. In contrast, basing a model

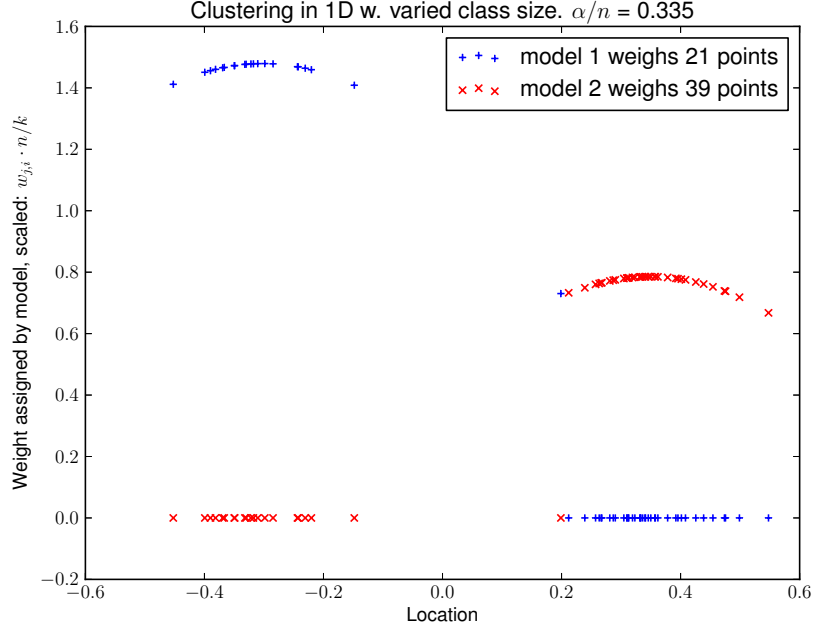

Figure 2.2: For each location (horizontal) of a data point, the vertical locations of corresponding markers gives the weights assigned by each model. The left cluster is half as populated as the right, thus must give weights about twice as large. Within each model, weights are affine in the loss (see Section 2.1), causing the concave parabolas. The gap allowed between the maximal weights of different models allows a point from the right cluster to be adopted by the left model, lowering overall penalty at a cost to weighted losses.

on very few points (mode II of sensitivity) should be avoided. If the $j$th model is fit to only $\gamma n k^{-1}$ points for $\gamma \ll 1$, the penalty from those points will be at least (approximately) $\alpha n^{-1} \cdot \gamma^{-1} k^{-1}$. We can make the first situation cheap and the second expensive (per model) in comparison to the empirical weighted loss term by choosing

$$\alpha n^{-1} \approx k^{-1} \sum_{j=1}^{k} \mathbf{w}_j^\top \mathbf{l}_j. \tag{2.4}$$

On the flip side, highly unbalanced classes in the data can be challenging to our approach. Consider the case where a model has low loss for fewer than $n/(2k)$ points: spreading its weight only over them can incur very high costs due to the regularization term, which might be lowered by including some higher-loss points that are indeed better explained by another model (see Figure 2.2 on page 5 for an illustration). This challenge might be solved by explicitly and separately estimating the relative frequencies of the classes, and penalizing deviations from the estimates rather than from equal frequencies, as is done in mixture models [6]; this is left for future study.

## 2.1 Two properties of Regularized Weighting

Two properties of our formulation result from an analysis (in Appendix A for lack of space) of a dual problem of the weight setting problem (2.3). These provide the basis for later theory by relating $\mathbf{v}$, losses and $\alpha$. The first illustrates the uniform control of $\mathbf{v}$:

**Lemma 1.** *Let all losses be in* $[0, B]$*, then in an optimal solution to (2.3), we have*
$$\|\mathbf{v} - \mathbf{u}\|_\infty \leq B/(2\alpha).$$

This strengthens the conclusion of (2.4): if outliers are present and $\alpha n^{-1} > 2B$ where $B$ bounds losses on all points including outliers, weights will be almost uniform (enabling mode I of sensi-

tivity). On the positive side, this lemma plays an important role in the generalization and iteration complexity results presented in the sequel. A more detailed view of $\mathbf{v}_i$ for individual points is provided by the second property.

By $P_C$ we denote the orthogonal projection mapping into a convex set $C$.

**Lemma 2.** *For an optimal solution to (2.3), there exists $t \in \mathbb{R}^k$ such that:*

$$\mathbf{v} = P_{\triangle^n} \left( \mathbf{u} - \min_j \left( \mathbf{l}_j - t_j \right) / (2\alpha) \right),$$

*where $\min_j$ should be read as operating element-wise, and in particular $\mathbf{w}_{j,i} > 0$ implies that $j$ minimizes the $i$th element.*

This establishes that average weight (when positive) is affine in the loss; the concave parabolas visible in Figure 2.2 on page 5 are an example. We also learn the role of $\alpha$ in solutions is determining the coefficient in the affine relation. Distinct $t$ allow for different densities of points around different models. One observation from this lemma is that if a particular model $j$ gives weight to some point $i$, then every point with lower loss $\ell(x_{i'}, m_j)$ under that model will receive at least that much weight. This property plays a key role in the proof of robustness to outliers in clustering.

## 2.2 An alternating optimization algorithm

The RW multiple model learning loss, like other MML losses, is not convex. However the weight setting problem (2.3) is convex when we fix the models, and an efficient procedure $\mathcal{A}$ is assumed for solving a weighted base learning problem for a model, supporting an alternating optimization approach, as in Algorithm 1; see Section 5 for further discussion.

---

**Data**: $X$
**Result**: The model-set $M$
$M \leftarrow initialModels\,(X)$;
**repeat**
    $M' \leftarrow M$;
    $W \leftarrow \arg\min_{W'} L_\alpha\,(X, M, W')$;
    $m_j \leftarrow \mathcal{A}\,(\mathbf{w}_j, X) \quad (\forall j \in [k])$;
**until** $L\,(X, M') - L\,(X, M) < \varepsilon$;

**Algorithm 1:** Alternating optimization for Regularized Weighting

---

# 3 Breakdown point in clustering

Our formulation allows a few difficult outliers to be ignored if the right models are found; does this happen in practice? Figure 2.1 on page 4 provides a positive example in regression clustering, and a more substantial empirical evaluation on subspace clustering is in Appendix B. In the particular case of clustering with the squared Euclidean loss, robustness benefits can be proved.

We use "breakdown point" – the standard robustness measure in the literature of *robust statistics* [2], see also [17, 18] and many others – to quantify the robustness property of the proposed formulation. The breakdown point of an estimator is the smallest fraction of bad observations that can cause the estimator to take arbitrarily aberrant values, i.e., the smallest fraction of outliers needed to *completely break* an estimator.

For the case of clustering with the squared Euclidean distance base loss, the min-loss approach corresponds to $k$-means clustering which is not robust in this sense; its breakdown point is 0. The non robustness of $k$-means has led to the development of many formulations of robust clustering, see a review by [14]. In contrast, we show that our joint loss yields an estimator that has a non-zero breakdown point, and is hence robust.

In general, a squared loss clustering formulation that assigns equal weight to different data points cannot be robust – as one data point tends to infinity so must at least one model. This applies to our model if $\alpha$ is allowed to tend to infinity. On the other hand if $\alpha$ is too low, it becomes possible

for each model to assign all of its weight to a single point, which may well be an outlier tending to infinity. Thus, it is well expected that the robustness result below requires $\alpha$ to belong to a data dependent range.

**Theorem 2.** *Let $\mathcal{X} = \mathcal{M}$ be a Euclidean space in which we perform clustering with the loss $\ell(x_i, m_j) = \|m_j - x_i\|^2$ and $k$ centers. Denote by $R$ the radius of any ball containing the inliers, and $\eta < k^{-2}/22$ the proportion of outliers allowed to be outside the ball. Denote also by $r$ a radius such that there exists $M' = \{m'_1, \cdots, m'_k\}$ such that each inlier is within a distance $r$ of some model $m'_j$ and each $m_j$ approximates (i.e., within a distance $r$) at least $n/(2k)$ inliers; this always holds for some $r \leq R$.*

*For any $\alpha \in n\left[r^2, 13R^2\right]$ let $(M, W)$ be minimizers of $L_\alpha(X, M, W)$. Then we have $\|m_j - x_i\|_2 \leq 6R$ for every model $m_j$ and inlier $x_i$.*

Theorem 2 shows that when the number of outliers is not too high, then the learned model, regardless of the magnitude of the outliers, is close to the inliers and hence cannot be arbitrarily bad. In particular, the theorem implies a non-zero breakdown point for any $\alpha > nr^2$; taking too high an $\alpha$ merely forces a larger but still finite $R$. If the inliers are amenable to balanced clustering so that $r \ll R$, the regime of non-zero breakdown is extended to smaller $\alpha$.

The proof follows three steps. First, due to the regularization term, for any model, the total weight on the few outliers is at most $1/3$. Second, an optimal model must thus be at least twice as close to the weighted average of its inlier as it is to the weighted average of its outliers. This step depends critically on squared Euclidean loss being used. Lastly, this gap in distances cannot be large in absolute terms, due to Lemma 2; an outlier that is much farther from the model than the inliers must receive weight zero. For the proof see Appendix C of the supplementary material.

## 4 Regularized Weighting formulation sample complexity

An important consideration in learning algorithms is controlling overfitting, in which a model is found that is appropriate for some data, rather than for the source that generates the data. The current formulation seems to be particularly vulnerable since it allows data to be ignored, in contrast to most generalization bounds that assume equal weight is given to all data.

Our loss $L_\alpha(X, M)$ differs from common losses in allowing data points to be differently weighted. Thus, to obtain the sample complexity of our formulation we need to bound the difference that a single sample can make to the loss. For a common empirical average loss this is bounded by $Bn^{-1}$ where $B$ is the maximal value of the non-negative loss on a single data point, and in our case by $B\|\mathbf{v}\|_\infty$, because if $X, X'$ differ only on the $i$th element, then:

$$|L_\alpha(X', M, W) - L_\alpha(X, M, W)| = \left|k^{-1}\sum_{j=1}^{k}\left(\mathbf{w}_{j,i}\left(\mathbf{l}_{j,i} - \mathbf{l}'_{j,i}\right)\right)\right| \leq Bk^{-1}\sum_{j=1}^{k}\mathbf{w}_{j,i} \leq B\mathbf{v}_i.$$

Whenever $W$ is optimal with respect to either $X$ or $X'$, Lemma 1 provides the necessary bound on $\|\mathbf{v}\|_\infty$. Along with covering numbers as defined next and standard arguments (found in the supplementary material), this bound on differences provides us with the desired generalization result.

**Definition 3** (Covering numbers for multiple models)**.** We shall endow $\mathcal{M}^k$ with the metric

$$d_\infty(M, M') = \max_{j\in[k]}\left\|\ell(\cdot, m_j) - \ell(\cdot, m'_j)\right\|_\infty$$

and define its *covering number* $N_\varepsilon(\mathcal{M}^k)$ as the minimal cardinality of a set $\mathcal{M}^k_\varepsilon$ such that $\mathcal{M}^k \subseteq \bigcup_{M\in\mathcal{M}^k_\varepsilon} B(M, \varepsilon)$.

The bound depends on an upper bound on base losses denoted $B$; this should be viewed as fixing a scale for the losses and is standard where losses are not naturally bounded (e.g., classical bounds on SVM kernel regression [19] use bounded kernels). Thus, we have the following generalization result, whose proof can be found in Appendix D of the supplementary material.

**Theorem 3.** *Let the base losses be bounded in the interval* $[0, B]$, *let* $\mathcal{M}^k$ *have covering numbers* $N_\varepsilon(\mathcal{M}^k) \le (C/\varepsilon)^{dk}$ *and let* $\gamma = nB/(2\alpha)$. *Then we have with probability at least* $1 - \exp\left\{ dk \log\left(\frac{2C}{\tau}\right) - \frac{2n\tau^2}{B^2(1+\gamma)^2} \right\}$:

$$\forall M \in \mathcal{M}^k \quad |L_\alpha(X, M) - \mathbb{E}_{X' \sim D^n} L_\alpha(X', M)| \le 3\tau.$$

## 5 The weight assignment optimization step

As is typical in multi-model learning, simultaneously optimizing the model and the association of the data (in our formulation, the weight) is computationally hard [20], thus Algorithm 1 alternates between optimizing the weight with the model fixed, and optimizing the model with the weights fixed. Thus we show how to efficiently solve a sequence of weight setting problems, minimizing $L_\alpha(X, M_i, W)$ over $W$, where $M_i$ typically converge.

We propose to solve each instance of weight setting using gradient methods, and in particular FISTA [21]. This has two advantages compared to Interior Point methods: First, the use of memory for gradient methods depends only linearly with respect to the dimension, which is $O(kn)$ in problem (2.3), allowing scaling to large data sets. Second, gradient methods have "warm start" properties: the number of iterations required is proportional to the distance between the initial and optimal solutions, which is useful both due to bounds on $\|\mathbf{v} - \mathbf{u}\|_\infty$ and when $M_i$ converge.

**Theorem 4.** *Given data and models* $(X, M)$ *there exists an algorithm that finds a weight matrix* $W$ *such that* $L_\alpha(X, M, W) - L_\alpha(X, M) \le \varepsilon$ *using* $O\left(\sqrt{k\alpha/\varepsilon}\right)$ *iterations, each costing* $O(kn)$ *time and memory. If* $\alpha \ge Bn/4$ *then* $O\left(k\sqrt{\alpha n^{-1}/\varepsilon}\right)$ *iterations suffice.*

The first bound might suggest that typical settings of $\alpha \propto n$ requires iterations to increase with the number of points $n$; the second bounds shows this is not always necessary.

This result can be realized by applying the algorithm FISTA, with a starting point $\mathbf{w}_j = \mathbf{u}$, with $2\alpha k^{-2}$ as a bound on the Lipschitz constant for the gradient. For the first bound we estimate the distance from $\mathbf{u}$ by the radius of the product of $k$ simplices; for the second we use Lemma 1 in Appendix E.

## 6 Conclusion

In this paper, we proposed and analyzed, from a general perspective, a new formulation for learning multiple models that explain well much of the data. This is based on associating to each model a regularized weight distribution over the data it explains well. A main advantage of the new formulation is its robustness to fat tailed noise and outliers: we demonstrated this empirically for regression clustering and subspace clustering tasks, and proved that for the important case of clustering, the proposed method has a non-trivial breakdown point, which is in sharp contrast to standard methods such as $k$-means. We further provided generalization bounds and explained an optimization procedure to solve the formulation in scale.

Our main motivation comes from the fast growing attention to analyzing data using multiple models, under the names of $k$-means clustering, subspace segmentation, and Gaussian mixture models, to list a few. While all these learning schemes share common properties, they are largely studied separately, partly because these problems come from different sub-fields of machine learning. We believe general methods with desirable properties such as generalization and robustness will supply ready tools for new applications using other model types.

**Acknowledgments**

H. Xu is partially supported by the Ministry of Education of Singapore through AcRF Tier Two grant R-265-000-443-112 and NUS startup grant R-265-000-384-133. This research was funded (in part) by the Intel Collaborative Research Institute for Computational Intelligence (ICRI-CI).

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
