[Supplementary Material]

# Supplementary Material

## A A dual problem: proof of Lemma 1 and Lemma 2

Our goal in this section is to gain an understanding of $\mathbf{v}$ in solutions of (2.3), culminating in the proof of Lemma 1. This understanding will come from considering optimization problems equivalent to a particular dual problem of the weight setting problem (2.3), and properties of the projection onto the probability simplex.

**Theorem 5.** *Denote*

$$g\left(\lambda\right) = \min_{\mathbf{v}\in\triangle^n}\left(\alpha\left\|\mathbf{u}-\mathbf{v}\right\|_2^2 - \lambda^\top\mathbf{v}\right) + k^{-1}\sum_{j=1}^k\min_i\left(\mathbf{l}_j+\lambda\right)_i.$$

*Then the problem* $\max_{\lambda\in\mathbb{R}^n} g\left(\lambda\right)$ *is a dual to (2.3). Strong duality holds.*

We proceed to prove the following relations between $\mathbf{v}$, $\lambda$ and the dual problem (optimal values denoted with $\cdot^*$):

1. $\arg\min_{\mathbf{v}\in\triangle^n}\left(\alpha\left\|\mathbf{u}-\mathbf{v}\right\|_2^2 - \lambda^\top\mathbf{v}\right) = P_{\triangle^n}\left(\mathbf{u}+\frac{\lambda}{2\alpha}\right)$, then $\mathbf{v}^* = P_{\triangle^n}\left(\mathbf{u}+\frac{\lambda^*}{2\alpha}\right)$.

2. $\nabla_\lambda\min_{\mathbf{v}\in\triangle^n}\left(\alpha\left\|\mathbf{u}-\mathbf{v}\right\|_2^2 - \lambda^\top\mathbf{v}\right) = -P_{\triangle^n}\left(\mathbf{u}+\frac{\lambda}{2\alpha}\right)$.

3. Denoting $\mathbf{t}_j = \min_i\left(\mathbf{l}_j+\lambda^*\right)_i$, an optimal $\lambda^*$ exists that can be written as $\lambda^* = \max_j\left(\mathbf{t}_j\mathbf{1}_n - \mathbf{l}_j\right)$. In particular $\max_{\lambda\in\mathbb{R}^n} g(\lambda) = \max_{\mathbf{t}\in\mathbb{R}^k} h(\mathbf{t})$ where

$$h(\mathbf{t}) = \min_{\mathbf{v}\in\triangle^n}\left(\alpha\left\|\mathbf{u}-\mathbf{v}\right\|_2^2 - \left(\max_j\left(t_j\mathbf{1}_n - \mathbf{l}_j\right)\right)^\top\mathbf{v}\right) + k^{-1}\sum_{j=1}^k t_j. \qquad (\text{A.1})$$

4. Invariance: $g(\lambda) = g\left(\lambda + a\cdot\mathbf{1}_n\right)$ and $h(\mathbf{t}) = h\left(\mathbf{t}+a\cdot\mathbf{1}_k\right)$.

5. Dual maximizers $\lambda^*$ derived from $\mathbf{t}$ fulfill: $\max_i\lambda_i^* - \min_i\lambda_i^* \le \max_{j,i}\mathbf{l}_{j,i} - \min_{j,i}\mathbf{l}_{j,i}$.

6. Norm of $\mathbf{u}-v^*$: $\left\|\mathbf{u}-v^*\right\|_\infty^2 \le n\left(\max\lambda^* - \min\lambda^*\right)^2/\left(16\alpha^2\right)$. If losses are in $[0,B]$ and $\alpha \ge nB/4$, then we have

$$\left\|\mathbf{v}^*\right\|_2^2 \le 2/n.$$

We begin with the dual problem, and point to further lemmas for its properties. Properties 1 and 2 are proved in Lemma 3. Properties 3, 5 and 6 are treated in the next subsection, and property 4 is easy to verify.

*Proof.* In the primal optimization problem (2.3) we maintain the simplex constraints on $\mathbf{w}_j$ and on $\mathbf{v}$ implicitly using the domain and can express the constraint $\frac{1}{k}\sum_{j=1}^k\mathbf{w}_{j,i} = \mathbf{v}_i$ (for $i\in[n]$) as $h_i\left(W,\mathbf{v}\right) = \left(k^{-1}\sum_{j=1}^k\mathbf{w}_j\right)_i - \mathbf{v}_i = 0$, with the corresponding variable $\lambda\in\mathbb{R}^n$.

Taking the Lagrangian with respect to $h_i$ we have for $\mathbf{v},\mathbf{w}_j\in\triangle^n$, $\lambda\in\mathbb{R}^n$:

$$L\left(\left(\mathbf{w}_j\right)_{j=1}^k,\mathbf{v},\lambda\right) = \alpha\left\|\mathbf{u}-\mathbf{v}\right\|_2^2 + k^{-1}\sum_{j=1}^k\mathbf{l}_j^\top\mathbf{w}_j + \lambda^\top\left(k^{-1}\left(\sum_{j=1}^k\mathbf{w}_j\right)-\mathbf{v}\right)$$

The dual objective is highly separable:

$$g(\lambda) = \min_{\mathbf{w}_j, \mathbf{v} \in \triangle^n} L\left((\mathbf{w}_j)_{j=1}^k, \mathbf{v}, \lambda\right)$$

$$= \min_{\mathbf{w}_j, \mathbf{v} \in \triangle^n} \alpha \|\mathbf{u} - \mathbf{v}\|_2^2 + k^{-1} \sum_{j=1}^k \mathbf{l}_j^\top \mathbf{w}_j + \sum_{i=1}^n \lambda_i h_i(W, \mathbf{v}) \qquad \text{(A.2)}$$

$$= \min_{v \in \triangle^n} \left(\alpha \|\mathbf{u} - \mathbf{v}\|_2^2 - \lambda^\top \mathbf{v}\right) + k^{-1} \sum_{j=1}^k \min_{\mathbf{w}_j \in \triangle^n} \left((\mathbf{l}_j + \lambda)^\top \mathbf{w}_j\right)$$

$$= \min_{v \in \triangle^n} \left(\alpha \|\mathbf{u} - \mathbf{v}\|_2^2 - \lambda^\top \mathbf{v}\right) + k^{-1} \sum_{j=1}^k \min_i (\mathbf{l}_j + \lambda)_i. \qquad \text{(A.3)}$$

Strong duality holds by Slater's condition, witnessed by $\mathbf{v} = \mathbf{w}_j = \mathbf{u}$. $\qquad \square$

**Lemma 3.**

$$\mathbf{v}^* = \arg\min_{v \in \triangle^n} \left(\alpha \|\mathbf{u} - \mathbf{v}\|_2^2 - \lambda^\top \mathbf{v}\right) = P_{\triangle^n}\left(\mathbf{u} + \frac{\lambda}{2\alpha}\right) \in -\partial_\lambda \min_{v \in \triangle^n} \left(\alpha \|\mathbf{u} - \mathbf{v}\|_2^2 - \lambda^\top \mathbf{v}\right).$$

Next we consider the minimization subproblem $\arg\min_{v \in \triangle^n} \left(\alpha \|\mathbf{u} - \mathbf{v}\|_2^2 - \lambda^\top \mathbf{v}\right)$. We note that minimizing a standard (isotropic) convex quadratic over a convex set is equivalent to projecting the unconstrained minimum onto the convex set under the $l_2$ norm. We note further that a minimum of convex functions at a point includes in its sub-differential any subgradients of the functions achieve the minimum at that point.

The consequence in our case is

$$\mathbf{v}^* = \arg\min_{v \in \triangle^n} \left(\alpha \|\mathbf{u} - \mathbf{v}\|_2^2 - \lambda^\top \mathbf{v}\right) = P_{\triangle^n}\left(\mathbf{u} + \frac{\lambda}{2\alpha}\right) \in -\partial_\lambda \min_{v \in \triangle^n} \left(\alpha \|\mathbf{u} - \mathbf{v}\|_2^2 - \lambda^\top \mathbf{v}\right).$$

Thus the computation of $g$ and its gradients is dominated by the second term $\sum_{j=1}^k \min_i (\mathbf{l}_j + \lambda)_i$.

### A.1 An equivalent problem

We introduce the slack variables $t \in \mathbb{R}^k$:

$$\max_\lambda \left(\min_{v \in \triangle^n} \left(\alpha \|\mathbf{u} - \mathbf{v}\|_2^2 - \lambda^\top v\right) + k^{-1} \sum_{j=1}^k \min_i (\mathbf{l}_j + \lambda)_i\right) \equiv$$

$$\begin{array}{cc} \max_{t \in \mathbb{R}^k, \lambda \in \mathbb{R}^n} & \min_{v \in \triangle^n} \left(\alpha \|\mathbf{u} - \mathbf{v}\|_2^2 - \lambda^\top v\right) + k^{-1} \sum_{j=1}^k t_j \\ \text{s.t.} & t_j \le \min_i (\mathbf{l}_j + \lambda)_i \end{array} \qquad \text{(A.4)}$$

**Lemma 4.** *Any optimal solutions of A.4 must fulfill* $t_j = \min_i (\mathbf{l}_j + \lambda)_i$.

Due to the term $\sum_{j=1}^k t_j$ and the constraint. So the constraint is sharp w.r.t. $t$. Another way to read the constraint is that for all $i, j$, $t_j \le (\mathbf{l}_{j,i} + \lambda_i) \iff t_j - \mathbf{l}_{j,i} \le \lambda_i$. So we can also ask is the constraint sharp w.r.t. $\lambda$?

**Lemma 5.** *For any* $\{\mathbf{l}_j\}_{j=1}^k, \alpha$, *there exists an optimal solution of Problem A.4 such that* $\lambda_i = \max_j t_j - \mathbf{l}_{j,i}$.

After the proof of this lemma, we assume without loss of generality that the optimal $\lambda^*$ we refer to are of this type.

*Proof.* As we noted before this lemma, $\lambda_i \geq \max_j t_j - \mathbf{l}_{j,i}$ is a direct consequence of the constraints, then we need prove only the other direction $\lambda_i \leq \max_j t_j - \mathbf{l}_{j,i} \iff (\exists j)\, \lambda_i \leq t_j - \mathbf{l}_{j,i}$.

Due to Lemma 4 we can assume without loss of generality that $t_j = \min_i (\mathbf{l}_j + \lambda)_i$, then we are considering maxima of

$$g(\lambda) = \min_{\mathbf{v} \in \triangle^n} \left( \alpha \left\| \mathbf{u} - \mathbf{v} \right\|_2^2 - \lambda^\top \mathbf{v} \right) + k^{-1} \sum_{j=1}^k \min_i (\mathbf{l}_j + \lambda)_i$$

If $i \notin \bigcup_{j \in [k]} \arg\min_i (\mathbf{l}_j + \lambda)_i$, then $\frac{\partial}{\partial \lambda_i} g(\lambda) = -\mathbf{v}_i \leq 0$, so that decreasing $\lambda_i$ can only improve the solution. $\qquad \square$

Now we prove Lemma 2.

*Proof.* The first part is implied by Lemma 5. Now recall that strong duality holds in Theorem 5. As a consequence, any solution $\left( (\mathbf{w}_j)_{j=1}^k, \mathbf{v}, \lambda^* \right)$ must minimize the Lagrangian w.r.t. $\lambda^*$ and the implicit constraints.

In particular, for any $j$, $\mathbf{w}_j \in \triangle n$ should minimize $k^{-1} (\mathbf{l}_j + \lambda^*)^\top w_j$, which occurs if the support of $\mathbf{w}_j$ is a subset of $I = \arg\min (\mathbf{l}_{j,i} + \lambda_i^*)$.

Recalling Lemma 4, on $i \in I$ we have $t_j = \mathbf{l}_{j,i} + \lambda_i$. Comparing to $\lambda_i = \min_j (t_j - \mathbf{l}_{j,i})$ which we assume due to 5 completes the proof. $\qquad \square$

**Corollary 1.** *Let losses be bounded in the interval $[b, B]$, then $\max_i \lambda_i - \min_i \lambda_i \leq B - b$.*

*Proof.* From Lemma 5 we have $\lambda_{\max} \leq t_{\max} - b$. Applying Lemma 4, we find $t_{\max} \leq B + \lambda_{\min}$. $\qquad \square$

## A.2  Generic bound on $\|\mathbf{u} - \mathbf{v}\|_\infty$

when losses are in $[b, B]$.

**Lemma 6.** *Let losses be bounded in $[b, B]$. Then the following always holds:*

$$\left| \mathbf{v}_i - n^{-1} \right| \leq (B - b) / (2\alpha)$$

*Proof.* From Corollary 1, we know that $\max_i \lambda_i - \min_i \lambda_i \leq B - b$, and $\mathbf{v} = P_{\triangle n}(\mathbf{u} + \lambda/2\alpha)$, so $\mathbf{v}_i = \left[ n^{-1} + \lambda_i/2\alpha + a \right]_+$ for some $a$. First we will show that $|\mathbf{v}_i - \mathbf{v}_{i'}| \leq (B - b)/(2\alpha)$. The bound clearly holds for $|\lambda_i/2\alpha - \lambda_{i'}/2\alpha|$, and adding $n^{-1} + a$ does not change this. Similarly, $[\cdot]_+$ is a contraction.

The maximal value for any $\mathbf{v}_i \in \triangle n$ under this constraint is achieved when $\mathbf{v}_1 = (B - b)/(2\alpha) + \mathbf{v}_i$ for every $i \neq 0$. Then from the simplex constraint, we have: $1 = n\mathbf{v}_1 - (n - 1)(B - b)/(2\alpha)$ thus

$$\mathbf{v}_1 = n^{-1} + (n - 1) n^{-1} (B - b) / (2\alpha).$$

A similar proof holds for the minimal value $\mathbf{v}_i$ can achieve. $\qquad \square$

# B  Robustness in subspace clustering

To demonstrate empirically the robustness properties of solutions to (2.2), we apply it to the problem of subspace clustering, on data well described by subspaces before we add fat tailed noise. We then report the median distance (MD) of data points from the models found; higher MDs correspond to models more strongly affected by the few high noise points.

We apply our algorithm to the better balanced 80 datasets in the Hopkins155 video motion segmentation database [22]. Each data-set in this database consists of a set of vectors, each vector originating from one of two or three objects moving in the scene. The vectors corresponding to each object are all approximately on a subspace of dimension at most 4 (known in advance). We preprocess each

data-set by PCA to a dimension equal to the sums of dimensions of the original subspaces. We seek subspaces of dimension greater than most original subspaces in that data-set.

Each of a Lloyd-like algorithm and Algorithm 1 are restarted 40 times with random subspaces. Values for $\alpha$ are chosen randomly according to $n \cdot 10^y$ with $y \sim Unif([-2,1])$. For each algorithm and data-set, we report the result with lowest MD, alongside that from the ground truth association. Our formulation produces lower MD quite consistently, especially in the higher noise setting.

Figure B.1: On the 80 most class-balanced of the Hopkins 155 subspace clustering datasets, Algorithm 1 achieves generally better approximation without outliers, and significantly better approximation when fat tailed noise is introduced, both compared to an algorithm based on Lloyd's and compared to subspaces fit according to the ground truth segmentation.

## C    Robustness: proof of Theorem 2

Throughout this section the data set $X$ is a union of outliers about which no assumptions are made except that they number exactly $\eta n$ and inliers.

### C.1    Loss of an outlier ignoring solution

We begin by calculating our loss when inliers behave nicely, and outliers are ignored. This will allow us to reject some undesired situations as being suboptimal in comparison.

Below we denote by $\phi^{-1}(x)$ the inverse image of $x$, that is the set $y : \phi(x) = y$, and by $|\cdot|$ the cardinality of a set.

**Lemma 7.** *We have that*
$$\min_M L_\alpha\left(X, M\right) \leq \alpha n^{-1} + r^2$$

*if the following conditions hold:*

1. *The inliers can be explained well: there exists a model-set $M = \{m_j\}_{j=1}^k \subset \mathcal{M}$ and a mapping $\phi : X \to M$ such that for every inlier $x_i$ we have $\ell\left(x_i, \phi\left(x_i\right)\right) \leq r^2$.*

2. *Each model explains sufficient inliers: $\left|\phi^{-1}\left(m_j\right)\right| \geq n/(2k)$.*

*Proof.* Take the solution $M$, and distribute the weights $\mathbf{w}_j$ for model $j$ uniformly over $D_j = \phi^{-1}\left(m_j\right)$.

First we bound the regularization term. The average vector $\mathbf{v} = \frac{1}{k}\sum_{j=1}^k w_j$ is a distribution, zero on $x_i \notin \bigcup_{j=1}^3 D_j$, so $|\mathbf{u}_i - \mathbf{v}_i| = n^{-1}$ there.

To prove $|\mathbf{u}_i - \mathbf{v}_i| \leq n^{-1}$ elsewhere, it is enough to show that $\mathbf{v}_i \leq 2n^{-1}$. Because $D_i$ are disjoint, $\mathbf{v}_i = k^{-1}|D_j|^{-1}$ for some particular $j$. Then it is enough that $k^{-1}|D_j|^{-1} \leq 2n^{-1} \iff n2^{-1}k^{-1} \leq |D_j|$, as we assume. Then $\alpha\|\mathbf{u} - \mathbf{v}\|_2^2 \leq \alpha\sum_{i=1}^n n^{-2} \leq \alpha n^{-1}$.

By construction, we give weight only where losses are bounded above by $r^2$, then the bound is preserved by the weighted and unweighted means. □

**Corollary 2.** *For $\alpha = b^2 n$ with $b \geq r$ under the conditions of the lemma above, the loss is upper bounded by $2b^2$.*

## C.2 Lower bound on loss when a model is far from data, and outliers are sufficiently few

**Lemma 8.** *Under the assumptions of Lemma 7 and $\alpha \geq n \cdot r^2$ and also $\eta < k^{-2}/22$, for any model $m_j$ in an optimal solution $M = \{m_j\}_{j=1}^k$, any $\eta n$ points receive a total weight of less than $1/3$ in $\mathbf{w}_j$.*

We show that the regularization term itself is sufficient to exclude such concentration of weight in the outliers.

*Proof.* In the regularization term, the part of the sum corresponding to outliers and $\mathbf{w}_j$ alone is minimized by distributing the total weight $p$ they are assigned from $\mathbf{w}_j$ uniformly among the (at most) $\eta n$ outliers. Then the loss is bounded below by

$$\alpha \sum_{i \in \text{outliers}} \left(\mathbf{u}_i - k^{-1}p/(\eta n)\right)^2 = \alpha(\eta n)\left(n^{-1} - pk^{-1}/(\eta n)\right)^2$$

$$= \alpha \eta n^{-1}\left(1 - pk^{-1}/\eta\right)^2$$

$$= \alpha n^{-1}\eta^{-1}\left(\eta - pk^{-1}\right)^2$$

Then if we show that $\eta^{-1}\left(\eta - pk^{-1}\right)^2 > 2$ we have $\alpha n^{-1}\eta^{-1}\left(\eta - pk^{-1}\right)^2 > 2\alpha n^{-1} \geq \alpha n^{-1} + r^2$ since $\alpha \geq n \cdot r^2$, then by Lemma 7 $M$ is suboptimal.

$$\eta^{-1}\left(\eta - pk^{-1}\right)^2 = \eta^{-1}\left(\eta^2 - 2\eta pk^{-1} + p^2 k^{-2}\right)$$

$$(\eta \geq 0) \geq \eta^{-1}p^2 k^{-2} - 2pk^{-1}$$

$$(\text{assumption about } \eta) \geq \left(k^{-2}/22\right)^{-1}p^2 k^{-2} - 2pk^{-1}$$

$$= 22p^2 - 2pk^{-1}$$

$$(k \geq 2) \geq 22p^2 - p$$

The quadratic is larger than 2 at $p = 1/3$ because $22/9 - 1/3 = 19/9 > 2$, and because it is positive and its derivative is positive there $22 \cdot 2p - 1 = 44/3 - 1 > 0$, and only increases with $p$. □

We proceed to the proof of Theorem 2.

The geometric idea behind the proof is that because the outliers have smaller total weight than the set of inliers, their weighted average will be farther than that of the inliers from every model. Being far, the outliers have high loss, so decreasing their weight improves $L_\alpha$ as long as $\alpha$ is not too high.

*Proof.* Without loss of generality, we may assume the ball containing the inliers is centered at the origin. The proof is by contraposition: we will assume a model at $\|m_j\| > 5R$ and show this implies $13R^2 < \alpha n^{-1}$. We will first estimate some losses and then pass to the corresponding weights.

The model $m_j$ has by Lemma 8 at least two thirds of weight at inliers denoted $i$. The average of inliers $p^i$ as weighted by $\mathbf{w}_j$ must be in $B(0, R)$, therefore at distance $\left\|m_j - p^i\right\| \geq \|m_j\| - \|p^i\| \geq 4R$, then the average of outliers $p^o$ as weighted by $\mathbf{w}_j$ must be at distance $\left\|p^o - m_j\right\| \geq 2\left\|p^i - m_j\right\| \geq \|p_i - m_j\| + 4R$, and at least one outlier $i'$ with positive weight must be at distance that is no smaller. With respect to any inlier $i$, $m_j$ has loss level $\|x_i - m_j\|^2 \leq \left(\left\|m_j - p^i\right\| + \left\|p^i - x_i\right\|\right)^2 \leq \left(\left\|m_j - p^i\right\| + 2R\right)^2$ by the triangle inequality.

By Lemma 2, $\mathbf{v} = \left[\mathbf{u} - \min_j \left(\mathbf{l}_j - t_j\right) / \left(2\alpha\right) + a\right]_+$, and $j$ is a minimizer at the outlier $i'$. This will help us lower bound $\mathbf{v}_i$ at inliers. Combined with the loss estimates above, we have:

Then

$$\min_{j'} \left(\mathbf{l}_{j',i} - t_{j'}\right) \leq \mathbf{l}_{j,i} - t_j \leq$$

$$\left(\left\|m_j - p^i\right\| + 2R\right)^2 - t_j \leq \left(\left\|m_j - p^i\right\| + 4R\right)^2 - t_j \leq$$
$$\mathbf{l}_{j,i'} - t_j = \min_{j'} \left(\mathbf{l}_{j',i'} - t_{j'}\right)$$

and in particular, estimating again:

$$28R^2 \leq -\min_{j'} \left(\mathbf{l}_{j',i} - t_{j'}\right) - \left(-\min_{j'} \left(\mathbf{l}_{j',i'} - t_{j'}\right)\right)$$

Then since $\mathbf{v}_{i'} > 0$, every inlier $i$ does receive weight under some model, and $\mathbf{v}_i \geq \mathbf{v}_{i'} + 14R^2/\alpha$. There are $n\left(1 - \eta\right)$ inliers, and we cannot have the inliers weight sum to more than 1, we must have $n\left(1 - \eta\right)14R^2/\alpha \leq 1$ which implies $13R^2 < \left(1 - \eta\right)14R^2 \leq \alpha n^{-1}$. □

## D  Sample complexity: proof of Theorem 3

*Proof.* By Lemma 1, $\|\mathbf{v}\|_\infty \leq n^{-1}\left(1 + \gamma\right)$. First we show that $L_\alpha\left(X, M\right)$ is difference bounded by $Bn^{-1}\left(1 + \gamma\right)$ with regard to its first argument.

Let $X, X'$ differ only at the $i$th element and let $M$ be fixed. Take $\left(\mathbf{w}_j\right)_{j=1}^k = W \in \arg\min_W \left(L_\alpha\left(X, M, W\right)\right)$ then

$$\left|L_\alpha\left(X', M, W\right) - L_\alpha\left(X, M\right)\right| = \left|k^{-1} \sum_{j=1}^k \left(\mathbf{w}_{j,i}\left(\mathbf{l}_{j,i} - \mathbf{l}'_{j,i}\right)\right)\right|$$

$$\leq Bk^{-1} \sum_{j=1}^k \mathbf{w}_{j,i}$$

$$\leq B\mathbf{v}_i$$

$$\leq Bn^{-1}\left(1 + \gamma\right).$$

Then by symmetry $\left|L_\alpha\left(X', M\right) - L_\alpha\left(X, M\right)\right| \leq Bn^{-1}\left(1 + \gamma\right)$ whenever $X, X'$ differ by at most one element.

Then by McDiarmid's inequality and using our bound on $\|\mathbf{v}\|$, we have for every fixed $M$:

$$P\left(\left|L_\alpha\left(X, M\right) - \mathbb{E}_{X' \sim D^n} L_\alpha\left(X', M\right)\right| > \tau\right) \leq 2e^{-\frac{2\tau^2}{\Sigma_{i=1}^n \left(Bn^{-1}\left(1+\gamma\right)\right)^2}}$$

$$= 2e^{-\frac{2n\tau^2}{B^2\left(1+\gamma\right)^2}}$$

By applying a union bound over the $\varepsilon$ covering $\mathcal{M}_\varepsilon^k$ of $\mathcal{M}^k$ we have

$$P\left(\exists M \in \mathcal{M}_\varepsilon^k \text{ such that } \left|L_\alpha\left(X, M\right) - \mathbb{E}_{X' \sim D^n} L_\alpha\left(X', M\right)\right| > \tau\right) \leq 2\left(\frac{C}{\tau}\right)^{dk} e^{-\frac{2n\tau^2}{B^2\left(1+\gamma\right)^2}}$$

$$= e^{\log 2 + dk\log\left(\frac{C}{\tau}\right) - \frac{2n\tau^2}{B^2\left(1+\gamma\right)^2}}$$

$$\leq e^{dk\log\left(\frac{2C}{\tau}\right) - \frac{2n\tau^2}{B^2\left(1+\gamma\right)^2}}$$

Then with probability at least $1 - \exp\left\{ dk \log\left(\frac{2C}{\tau}\right) - \frac{2n\tau^2}{B^2(1+\gamma)^2}\right\}$, we have that for every $M \in \mathcal{M}^k$, and let $M^c$ be the nearest element to $M$ in a $\tau$ cover of $\mathcal{M}^k$,

$$
\begin{aligned}
|L_\alpha(X, M) - \mathbb{E}_{X' \sim D^n} L_\alpha(X', M)| &\leq |L_\alpha(X, M) - L_\alpha(X, M^c)| \\
&\quad + |L_\alpha(X, M^c) - \mathbb{E}_{X' \sim D^n} L_\alpha(X', M^c)| \\
&\quad + |\mathbb{E}_{X' \sim D^n} L_\alpha(X', M^c) - \mathbb{E}_{X' \sim D^n} L_\alpha(X', M)| \\
&\leq 3\tau,
\end{aligned}
$$

using Lemma 9. $\qquad\square$

The following lemma relates the cover numbers discussed to our loss.

**Lemma 9.** *For every $\alpha$,*

$$
\|L_\alpha(\cdot, M) - L_\alpha(\cdot, M')\|_\infty \leq d_\infty(M, M').
$$

*Proof.* Let $X \in \mathcal{X}^n$ and $W \in \arg\min_W (L_\alpha(X, M, W))$. By Holder's inequality we have $\left|\mathbf{w}_j^\top (\mathbf{l} - \mathbf{l}')\right| < \varepsilon$ because $\mathbf{w}_j \in \triangle^n$. In particular, $L_\alpha(X, M) + \varepsilon \geq L_\alpha(X, M', W) \geq L_\alpha(X, M')$. This holds for every $X \in \mathcal{X}^n$ and also with $M, M'$ exchanged. $\qquad\square$

# E   Computational complexity: proof of Theorem 4

*Proof.* The computational requirements of a FISTA iteration consist of calculating a gradient and projecting a point onto the problem constraints. For these purposes we minimize (2.2) over $W \in (\triangle^n)^k$. The constraints are separable in the sense that we can perform $k$ projections into a simplex of dimension $n$, which require $O(n)$ time each[23]. The computation of the gradient also has time and space cost of $O(kn)$ (see Lemma 10 for details). We now prove the bounds on number of iterations needed.

We use a bound from [21]. Using $q$ to denote the number of iterations, and choosing their parameters $\alpha = \eta = 2$ it is enough to take the number of steps to be

$$
q \geq \|x_0 - x^*\|_2^2 \sqrt{\frac{L}{\varepsilon}}
$$

where $x_0$ is that starting point, $x^*$ is any optimal point and the function $f$ being minimized to within $\varepsilon$ of its optimal value and assuming $f$ has an $L$ Lipschitz gradient.

In the current problem, we take $x_0 = U \in \mathbb{R}^{k \times n}$ to be the initial weight matrix with $\mathbf{u}$ in each row, the gradient Lipschitz constant $L$ is the weight parameter $\alpha$, then the bound is:

$$
q \geq \|U - W^*\|_2 \sqrt{\frac{\alpha}{\varepsilon}}.
$$

To obtain the first result in Theorem 4, we simply note that $\|U - W^*\|_2^2 = \sum_{j=1}^{k} \sum_{i=1}^{n} \left(\mathbf{u}_i - \mathbf{w}_{i,j}^*\right)^2 \leq \sum_{j=1}^{k} \|\mathbf{u} - e_1\|_2^2 \leq k$.

To obtain the second result we bound:

$$
\begin{aligned}
\|U - W^*\|_2^2 &= \sum_{j=1}^{k} \|\mathbf{u} - \mathbf{w}_j^*\|_2^2 \\
&= \sum_{j=1}^{k} \left( \|\mathbf{w}_j^*\|_2^2 - \|\mathbf{u}\|_2^2 \right) \\
&\leq \sum_{j=1}^{k} \|\mathbf{w}_j^*\|_2^2 \\
&= \sum_{i=1}^{n} \sum_{j=1}^{k} \left(\mathbf{w}_{j,i}^*\right)^2.
\end{aligned}
$$

Given that $\sum_{j=1}^{k} \mathbf{w}_{j,i} = k\mathbf{v}_i$ and $\mathbf{w}_{j,i} \geq 0$, the $l_2$ norm of the column $\mathbf{w}_{\cdot,i}$ is bounded above by $k\mathbf{v}_i$. Squaring both sides we have $k^2 \mathbf{v}_i^2 \geq \sum_{j=1}^{k} \mathbf{w}_{j,i}^2$, then $\sum_{i=1}^{n} \sum_{j=1}^{k} \left(\mathbf{w}_{j,i}^*\right)^2 \leq k^2 \sum_{i=1}^{n} \mathbf{v}_i^2$. We apply Lemma 1 to $\alpha = \gamma n B/2$ for $\gamma > 1/2$ and find $\mathbf{v}_i \leq n^{-1}\left(1 + \gamma^{-1}\right)$, so the iteration bound is $\|U - W\|_2 \sqrt{\alpha/\varepsilon} \leq 3k\sqrt{B\gamma/\varepsilon}$. Then $\|U - W^*\|_2 \leq 3k\sqrt{n^{-1}}$. Substituting that bound, using the assumption on $\gamma$, and substituting $\gamma$ back into the iteration bound formula gives $\|U - W\|_2 \sqrt{\alpha/\varepsilon} \leq 3k\sqrt{\alpha n^{-1}/\varepsilon}$ as wanted. $\qquad \square$

**Lemma 10.** $\triangledown \left(\alpha \|\mathbf{u} - \mathbf{v}\|_2^2 + k^{-1} \sum_{j=1}^{k} \mathbf{l}_j^\top \mathbf{w}_j\right)$ *can be computed in* $O(kn)$ *time and space.*

*Proof.* Denote $f(x) = \alpha \|x\|^2$, $A = \begin{pmatrix} I_n & I_n & \cdots & I_n \end{pmatrix}/k \in \mathbb{R}^{n \times nk}$ then we wish to find the gradient of $f\left(A\mathbf{w}^f - \mathbf{u}\right)$, where $\mathbf{w}^f$ is the flat concatenation of the vectors $\mathbf{w}_j$.

Then $\triangledown \left(f\left(A\mathbf{w}^f - u\right)\right) = A^\top \triangledown f\left(A\mathbf{w}^f - u\right) = A^\top \left(2\alpha \left(A\mathbf{w}^f - u\right)\right)$, and

$$\triangledown \left(\alpha \|\mathbf{u} - \mathbf{v}\|_2^2 + k^{-1} \sum_{j=1}^{k} \mathbf{l}_j^\top \mathbf{w}_j\right) = A^\top \left(2\alpha \left(A\mathbf{w}^f - u\right)\right) + k^{-1}\mathbf{l}^f$$

where $\mathbf{l}^f$ is to $(\mathbf{l}_j)_{i=1}^{k}$ as $\mathbf{w}^f$ is to $(\mathbf{w}_j)_{j=1}^{k}$.

The non trivial part is $A^\top \left(2\alpha \left(A\mathbf{w}^f - u\right)\right)$. Note that $A\mathbf{w}^f = \mathbf{v} = k^{-1} \sum_{j=1}^{k} \mathbf{w}_j$, computable in $O(kn)$. Given the vector $b = (2\alpha (\mathbf{v} - u)) \in \mathbb{R}^n$, $A^\top b$ can be computed by the concatenation of $k$ copies of $b/k$ requiring $O(kn)$ time. $\qquad \square$