[Reviews · NeurIPS 2013]

Submitted by Assigned_Reviewer_1

The authors propose in this paper a regularized weighting technique for multi-model learning. Multi-model learning consists in fitting a "mixture" of simple models to a data set in a quite generic sense where each model is associated to a convex loss and the mixture is optimized with respect to both the model parameters and their weights (each model is associated to a set of weights that describe how much the model claims to explain each data point). This model generalizes several classical approaches such as k-means and classical probabilistic mixture modeling. The authors propose to penalize the weights of each model so as to favor uniform weight distributions as a way to bring robustness of the mixture with respect to outliers. The paper contains several theoretical results and some limited experimental validation.

The paper is interesting and well written. In particular, the generality of the approach is very well presented. The authors also managed to pack a nice collection of results in a limited space. They pushed all the proofs in supplementary material, but nevertheless, the paper feels quite self contained. Limitations of the technique are also clearly presented (see Figure 2.2 for instance).

My overall impression is therefore very positive. That said, the paper is not without flaws. A very minor one concerns notations:
- \Delta^n is introduced a bit too late (in footnote 1 which falls on page 3 while the notation is used on page 2).
- P_C (the orthogonal projection operator) is defined only in the supplementary material while it is used in Lemma 2 (in the s.m. the definition is also a bit too delayed).
- while the gamma parameter used in the discussion below definition 2 is indirectly defined, I think the discussion would be clearer by being more explicit (for instance by saying that the proportion of zero weights (i.e., ignored data points) should be between this and that).

While the overall paper is very clear, Section 3.1 is a bit too cryptic and lacks too much details: how was the noise generated? what is the value of alpha? what is the effect of said value on the performances? does MAD in the legend stands for MD in the text? if the base learning is PCA, is there any dimensionality reduction done? if yes how? All in one, this unique experimental evaluation raises more questions that it brings answers or insights on the proposed method.

Coming back to the choice of alpha, it is obvious, as discussed in Section 2 and 3.2, that its actual value will have at least some impact on the performances of the final model. And of course, any practitioner would want to optimize said alpha. Yet the experimental section does not show the effects of alpha (whose value is not specified) and the authors offer almost no clue on how to tune it. This is quite a shortcoming in the paper, in my opinion. As alpha appears explicitly in Theorem 3, it is quite clear that even on a theoretical point of view the parameter has some complex effect on the results.

Another point is the upper bound assumption (B everywhere). I'm familiar with clipping techniques used in rather old papers (for instance the Zeger and Lugosi's 1995 paper, see http://www.code.ucsd.edu/~zeger/publications/journals/LuZe95-IT-Nonparametric/LuZe95-IT-Nonparametric.pdf) but I'm not sure how to understand the assumption in the present context. In particular, it seems too me that the very idea of fighting arbitrary outliers might prevent this assumption to be true in practice. In other words, moving from Theorem 3 to a more general one which applies when the bound assumption is not valid (as in done in the paper referenced above) does not seem completely obvious to me. I might be missing something that should then be emphasized more in the paper.

A final point is that despite the (short) Section 1.1, the discussion on related works seem quite limited. In particular, mixture models can accommodate many types of reasonable outliers by modeling them explicitly, for instance via some Student component instead of Gaussian ones, or via some explicit noisy components added to the mixture. Bayesian priors can be included into the mix to avoid some strong effects of very badly behaving outliers. I'm not saying that such an approach might lead to non zero breakdown point, but in practice, such variants do work very well.
Summary: A very complete paper on a new way of mixing simple models in a regularized way that brings guaranteed robustness against outliers which lacks only a better experimental evaluation. It would be however quite impossible to pack this evaluation into the paper if all theoretical results were to be kept.

Submitted by Assigned_Reviewer_3

This paper proposes a general formulation of multiple model learning, which introduces the weight of each data point with a regularization term and is robust to outliers. This paper provides theoretical support of the formulation with some empirical results.

Quality, clarity, and significance:
Although this paper contains some interesting results, there are many unclear descriptions in both texts and mathematical notations, which significantly deteriorate the quality and the significance of this paper.

- Definition 1 is not clear. Why the tuple (actually, it's a triplet) with two sets and one function is called a "problem"? I cannot see any problem from here. I think the problem is minimizing the weighted loss.
- l.109: What is the bold style m ? I think it's a vector of m_j from j=1 to k, but such a vector is defined as M in l.104.
- In Example 1, it is better to show that for each example what X, M, and l are.
- l.147: Why an outlier tends to infinity?
- Lemma 2. What is the definition of P_{\Delta^n} ?
- In 3.1 (empirical results), what is the actual alpha and how to set it?
As the authors say in l.73, this alpha should be set appropriately and I think this is important for this formulation. But how? For example, cross-validation cannot be used in unsupervised learning. So some strategy to set alpha is needed and it is also valuable to analyze the sensitivity with respect to changes in alpha.
- In Figure 3.1, I do not understand "Dataset" of x-axis. Why RW MAD monotonically increases if datasets change?
- l.320: It is better to state the mathematical definition of the breakdown point.
- l.336: In l(m_j, x_i), m_j and x_j are opposite (see Definition 1).
- l.337: It would be better for readability to add some intuitive explanation of the number "22".
- l.375: What is l(\cdot, m_j)? I think some x is needed for the place \cdot to define this value.

Originality:
This paper presents a generalization of multi-model learning with some regularization term and the originality is not so high. But for me it is ok since this approach is important for development of this area.
Summary: This paper contains interesting theoretical results, but the description is not sophisticated enough and empirical evaluation is not sufficient.

Submitted by Assigned_Reviewer_4

Multiple model learning is a generalization of clustering, and in this framework cluster centers can be learning models.
For each learning model, weighted data are assigned, and the weights averaged over learning models are restricted uniform weights with l2 regularization.
Thanks to this regularization term, a simple alternating optimization algorithm is derived.
Also some theoretical bounds of performance are obtained, which support robustness to outliers.

The motivation is clearly stated.
Theoretical analysis looks mathematically sound.
(some notations such as P_delta is defined in appendix,
but not defined in the main body. therefore the authors should carefully check the main manuscript is self-contained.)
Each set of weighted data can be related with (empirical) distribution, so it might be nice to discuss the properties of regularization from the viewpoint of probabilistic mixture models, not only from optimization perspective.
Summary: Based on l2 regularization of average weights of data, a new method of multiple model learning is proposed.
Numerical experiments support its efficiency and good performance.

Submitted by Assigned_Reviewer_5

This work proposes a new robust method and a new unified framework
for some learning task generalizing clustering models. It
allows to robustly deal with problems such as clustering, subspace
clustering or multiple regression by encompassing weights on the data
samples.
To better handle outliers, probability distributions over the data
(in classical settings Dirac distributions are mostly considered)
are attached to each "model". Moreover, the distributions are obtained
by minimizing a trade-off between two terms.
The first one is the (Euclidean)
distance between the average weight distribution to the uniform distribution.
It aims at producing spread out weights inside each
model (it is a regularization term).
The second one is a weighted loss function that take into account the
importance of each samples (it is a data fitting term).

Globally the notation, formulae and mathematics details
are really loose. The English level is also rather poor.
Worst, very often the clarity of the paper is at stake.
Though the motivation and the method proposed are interesting,
I feel that this version of the paper is rather a preliminary investigation,
than a camera-ready paper. More work is needed to make it clear.

For instance, to clarify their point, the authors should
provide, with careful details, what the proposed method does in the simplest
case of their framework, i.e., in the context of clustering (with possibly experiments
in the presence of outliers).
This could be better developed by moving Section 2.1 into the Appendix.

The experiment section is even worse than the theoretical part.
Not enough details are provided to understand the figures, and section
3.1 is particularly uninformative in term of practical performance.
Can the author compare with other methods for the specific task proposed,
as, for instance, for the clustering task mentioned earlier.

Last but not least, a longer discussion on the way the trade-off parameter
\alpha is chosen (both in theory and in practice) should be given.


Additional line by line comments:

l086: other generalizations of the the k-means algorithm could be referenced
here as the framework proposed by

Banerjee et al. "Clustering with Bregman divergences", JMLR, 2005.

l103--106: the notations are not introduced before they are used: X, M, Delta_n^k, etc.

l109: \Delta^{n 1} is not consistent with the notation defined just before.

l147: a a

l161: the footnote should be given on the previous page.

l185: mention that u=(1/n, ... , 1/n) (if I understand correctly)

l186: indices in the sum are not consistent...

l194: is the optimization over W or over one single w_j. Please remove
this ambiguity.

l200: so why using an unusual closeness index? can you motivate your choice?

l207: "average penalty" what is this referring to precisely?

l232: " has 1/3 the point"? I don't understand this sentence, neither do I get the
figure.

l254: P_\Delta^n: is never defined (only afterwards in the Appendix!!!).

l299-314: What is MAD referring to in the legend? it is nowhere defined before.

l368: the rhs depends on i, the lhs does not. Please correct.

l402: requires->required

l403: converge->converges

l457: Candes -> Cand\`es

l468: what is the nature of this work?article, conference, etc...

l553: "with the corresponding variable ...": this sentence is unclear

l778: is the equality true for all U? It seems false without more assumptions
that should be reminded.


Summary: This papers proposes a general robust formulation for handling multiple model learning (e.g. extending clustering algorithms) by considering optimized data distributions.
Author Feedback

Author rebuttal: We thank the reviewers.

We lacked space; this impacted the completeness of Section 3.1 and indirectly, through some editing errors (easily fixed), has also affected clarity in other sections. In most of them (like the projection operator) the reviewers pointed out the clear fix. We clarify below others, and answer questions about the paper hoping to make its contributions easier to see.

- How to choose alpha, theory and practice.
We review the theory first: If alpha > 2*nB by Lemma 1 the method will weight all points almost equally, which is not useful. And the sample complexity (in Theorem 3) scales as (1+B/(2\alpha/n)^2, so that alpha << 0.5*nB is also possibly prohibitive, but this lower bound on alpha can easily be overly pessimistic.
Practically, one might choose alpha experimentally from among a set of candidates by using a measure of quality that is independent of alpha (the role MD plays in Section 3.1, see below). Based on the observations above, I would choose alpha from a geometrically decreasing set beginning with 2nB. A sufficiently low alpha will lead to the weighted empirical loss term being zero (by overfitting on a few points and ignoring the rest), providing a natural and conservatively late stopping point for the search.

- Reviewer 1 asked about the interpretation of B as an assumption of boundedness in the context of arbitrary outliers.
We agree: the generalization bounds apply in the bounded losses case, which is distinct from the arbitrary outliers case in which we show robustness results. We believe it may be possible to use robustness results (where they apply) to weaken the assumption for generalization bounds to bound loss on *inliers* only, and over a smaller class of "reasonable" modelsets, but this work is ongoing. A truncation approach as in Lugosi and Zeger may also be applicable (after adaptation for weighting data) to the constrained variant for our penalized problem, but we do not yet know; thank you for this suggestion.

- Reviewer 5 asked about the behavior of our formulation in a simple case of clustering.
Though possibly the caption is not explicit enough, this is provided in Figure 2.2, which shows the behavior of our formulation on a very simple problem: clustering in 1 dimension. Every data point is shown twice at its corresponding location on the horizontal axis: the vertical location of each appearance gives the weight assigned to it by a different model. Because weight decreases affinely with loss within a model (in Lemma 2 we see that -(2/alpha)^-1 is the slope in this relation), and we use a squared distance loss, the positive weights within a model plot are determined by a concave quadratic function of location, whose peak indicates that model's center. Any single outlier far enough that both quadratics are negative at its location would receive weight zero, and thus be ignored. Lower alpha would cause a faster decay of weights with distance; this could mitigate the class imbalance problem, ignore some inliers, and for very low values cause estimation to break down entirely. While space considerations exclude using multiple plots, we could add some of this description.

- Reviewers 1, 3 and 5 asked about the experimental section, Figure 3.1 and alpha in experiments.
Figure 3.1 shows the losses from applying our algorithm, a Lloyds based variant and using the ground truth, each to many different datasets arising from distinct video segments. The loss reported is the median distance (MD) of data points from the nearest subspace found by the corresponding algorithm. MD is also used to choose the best of all random restarts of each algorithm. MD is independent of alpha, therefore can be used to choose it; we perform the restarts of the proposed algorithm with alpha chosen randomly over a range covering three orders of magnitude. Thus, no extra computational resources are assigned in this experiment to optimizing over alpha, and selecting a very precise value of alpha does not appear to be necessary to manifest the improved MD.
The increasing RW MD visible in the plot is a consequence of the order we chose for readability, not indicative of any underlying phenomenon; the datasets vary significantly in the scale of the MD achievable, so that arbitrary orderings of the datasets make the graph difficult to read.
MAD is indeed a typo, should be MD.
We used PCA to preprocess the data before applying any of the algorithms, the dimension was chosen to keep the first percentile of eigenvalues in each dataset.

- Reviewer 5 asked about the motivation in our choice (l200) of the Euclidean norm to compare weight distributions.
In l199 we state the goal of avoiding high concentration, which is formally justified by the Theorems that result; all of them stem from Lemmas 1 and 2, where Lemma 1 exactly means that high concentration is avoided for sufficiently high alpha.
That high concentration is made expensive by Euclidean penalization is illustrated less formally by some representative calculations in the next paragraph; maybe this should be made more explicit in l201 by saying instead "Under the Euclidean penalization term, the penalty for a data point given weight that is higher by a factor of h>>1 than the natural 1/n is approximately h^2 so that such concentration is made expensive. This is illustrated by a few examples in the next paragraph."

- In l109, the 1 is a footnote mark over \Delta^{n}, not a new notation.
- In l368, the rhs seems to depend on i while the lhs does not. In noting X,X' differ on one element, we should have written it is the ith one.
- In l778, a specific matrix U is used. It is defined in l764.